# Catalytic thiolation-depolymerization-like decomposition of oxyphenylene-type super engineering plastics via selective carbon–oxygen main chain cleavages
Yasunori Minami [1,2] ✉, Sae Imamura[1], Nao Matsuyama[1], Yumiko Nakajima [1] & Masaru Yoshida[1]

As the effective use of carbon resources has become a pressing societal issue, the importance of chemical recycling of plastics has increased. The catalytic chemical decomposition for plastics is a promising approach for creating valuable products under efficient and mild conditions. Although several commodity and engineering plastics have been applied, the decompositions of stable resins composed of strong main chains such as polyamides, thermoset resins, and super engineering plastics are underdeveloped. Especially, super engineering plastics that have high heat resistance, chemical resistance, and low solubility are nearly unexplored. In addition, many super engineering plastics are composed of robust aromatic ethers, which are difficult to cleave. Herein, we report the catalytic depolymerization-like chemical decomposition of oxyphenylene-based super engineering plastics such as polyetheretherketone and polysulfone using thiols via selective carbon–oxygen main chain cleavage to form electron-deficient arenes with sulfur functional groups and bisphenols. The catalyst combination of a bulky phosphazene base $P_4$-$t$Bu with inorganic bases such as tripotassium phosphate enabled smooth decomposition. This method could be utilized with carbon- or glass fiber-enforced polyetheretherketone materials and a consumer resin. The sulfur functional groups in one product could be transformed to amino and sulfonium groups and fluorine by using suitable catalysts.

Organic materials and products, from commodity plastics to engineering plastics and stable super engineering plastics are indispensable for society and are utilized in a variety of fields from general-purpose products to advanced materials. However, since the organic resources that comprise them are naturally finite, future societies will be required to reuse and recycle them once consumed, rather than simply dispose of them. One of the methodologies to achieve this goal is chemical recycling, i.e., the conversion of organic products into raw materials by means of organic reactions[1–14]. In this scenario, gasification chemical recycling of waste plastics to produce methanol, propylene, olefins, and so on is a promising method. However, gasification requires high-temperature conditions, and the resulting products must be converted back into organic raw substrates. Thus, chemical decomposition methodologies that convert plastics directly into raw organic compounds such as monomers at lower temperatures are becoming

increasingly important. Especially, plastics and polymers having relatively cleavable main chains such as an ester group are useful for this purpose and are being developed. For example, chemical recycling of polyethylene terephthalate (PET) has been extensively studied, giving usable low-weight molecules[15–20].

As mentioned above, many studies have developed the methodologies of the chemical decomposition of various resins, and recently, the focus is on catalytic decomposition for highly stable resins composed of strong main chains such as polyamides, polyurethanes, polyureas, thermoset resins, and super engineering plastics. For example, Nylon-6 was found to undergo decomposition in the presence of a dimethylaminopyridine[21–24] or lanthanide[25] catalyst to form ε-caprolactam. Catalytic hydrogenolysis was applicable to the decomposition of polyamides to produce amino alcohols[26,27]. Polyurethanes[28,29] and polyureas[30,31] were also subjected to

[1]Interdisciplinary Research Center for Catalytic Chemistry (IRC3), National Institute of Advanced Industrial Science and Technology (AIST), Tsukuba Central 5, 1-1-1 Higashi, Tsukuba, Ibaraki 305-8565, Japan. [2]PRESTO, Japan Science and Technology Agency (JST), 1-1-1 Higashi, Tsukuba, Ibaraki 305-8565, Japan. ✉e-mail: yasu-minami@aist.go.jp

catalytic hydrogenation to afford anilines, polyols, and amines. Decomposition of epoxy resins was developed using catalytic main-chain cleavage to provide the corresponding monomers[32–34]. Thus, the catalytic approach has the potential to achieve the decomposition of such stable resins to form useful low-weight molecules such as monomers. Among these stable resins, super engineering plastics are known for their excellent stability such as heat resistance and chemical resistance. Based on their high stability, these resins are indispensable to industries such as the automotive medical, aerospace, and other industries. However, catalytic decomposition of super engineering plastics remains nearly unexplored. A few catalytic decompositions of polyphenylenesulfide (PPS) composed of phenyl–sulfur bonds were reported to give low-molecular-weight molecules such as 1,4-dicyclopentylthiobenzene, benzene, and 1,4-dicyanobenzene (Fig. 1b)[35–38]. This scarcity of reports emphasizes the difficulty of catalytic decomposition of super engineering plastics. In addition, many super engineering plastics are composed of stable aromatic ethers, which are not easily cleaved.

Recently, we demonstrated that thiolate reagents are highly effective for the depolymerization-like chemical decomposition of PEEK using sulfur nucleophiles, giving monomer-like products, dithiofunctionalized benzophenones and hydroquinone (Fig. 1c)[39]. The electron-deficient carbonyl group in the PEEK main chain enhances the reactivity of the carbon-oxygen bond at the *para* position such that the highly nucleophilic thiolate reagents cleave this bond selectively. We applied this system to the chemical decomposition of PSU, PESU, and PEEK using stoichiometric amounts of CsOH·H$_2$O and CaH$_2$ to form the corresponding bisphenols[40]. We expected that these stoichiometric methods have the potential to be applied to base-catalyzed chemical decomposition of various super engineering plastics. Since the previous reactions proceeded smoothly under a moderate reaction

temperature (150 °C), the proposed catalytic strategy is expected to enable equally mild transformation to provide monomer-like products in high yields. Herein, we report the catalytic depolymerization-like chemical decomposition of oxyphenylene-based super engineering plastics using thiols to form monomer-like products, dithiofunctionalized arenes, and bisphenols (Fig. 1d). This method was applicable to PEEK, PSU, PPSU, and PEI. Inorganic bases and phosphazene bases were effective catalysts for this decomposition. Since the sulfur functional group acts as a leaving group for the substitution reaction under appropriate conditions[41,42], the produced dithiofunctionalized arenes could be converted into sulfonium cations followed by fluorination or aryloxylation reactions.

## Results and Discussion
### Optimization of the reaction conditions

We examined the chemical decomposition of insoluble polyetheretherketone (PEEK) powder ($M_w$ ~ 20800 and $M_n$ ~ 10300 as catalog specifications) (**1**) with 2-ethyl-1-hexanethiol (**2a**) (2 equiv. relative to monomer unit) in 1,3-dimethyl-2-imidazolidinone (DMI) under various conditions (Table 1). The decomposition was first performed using KOH, K$_3$PO$_4$, KO$t$Bu, and Cs$_2$CO$_3$ as catalysts (10 mol% relative to monomer unit) at 150 °C to form the corresponding decomposed products, dithiobenzophenone **4a**, 1,4-hydroquinone (**5**) and a benzophenone-hydroquinone-type dimer intermediate **3** (Table 1, Entries 1-4). The use of Cs$_2$CO$_3$ was especially effective to form the final decomposition monomers, **4a** and **5**, in good yields (Table 1, Entry 4), indicating that large counter cation sizes as well as basicity promote the decomposition. Encouraged by these results, we expected that bulky and strongly basic organic phosphazene bases such as P$_4$-$t$Bu (p$K_{BH+}$ 30.25 in dimethylsulfoxide (DMSO))[43–48] would be promising catalysts for this decomposition (Fig. 2), which

a)

Polyetheretherketone (PEEK)  Polysulfone (PSU)  Polyphenylsulfone (PPSU)

Polyetherethersulfone (PEES)  Polyethersulfone (PESU)  Polyetherimide (PEI)

b) PPS → Na$_2$S·9H$_2$O / K$_2$CO$_3$ then aq. HCl → m = 2~6 / reactant Pd cat. → R = S-cyclopentyl, H, CN

c) PEEK → PhCH$_2$CH$_2$SH / NaO$t$Bu then R−X → 

d) Y: carbonyl, sulfonyl, imide → R−SH catalyst → dithiofunctionalized arenes (Conversion of SR to (SRR)$^+$, F, OAr was possible.) + bisphenols

**Fig. 1 | Chemical decomposition of super engineering plastics. a** Examples of super engineering plastics. **b** Decomposition of PPS. **c** Previous work: PEEK decomposition using sulfur nucleophiles. **d** This work: Catalytic depolymerization-like chemical decomposition using thiols to afford dithiofunctionalized arenes and bisphenols. R–SH, organic thiol. R–X, organic halide. Ar, aryl.

**Table 1 | Optimization of catalytic chemical decomposition to form monomer-like products[a]**

| Entry | 2a (equiv.) | Catalyst | Solvent | Temp. (°C) | 3 (%) | 4a (%) | 5 (%) |
|---|---|---|---|---|---|---|---|
| 1 | 2 | KOH | DMI | 150 | 17 | 41 | 41 |
| 2 | 2 | $K_3PO_4$ | DMI | 150 | 12 | 54 | 56 |
| 3 | 2 | KOtBu | DMI | 150 | 12 | 62 | 62 |
| 4 | 2 | $Cs_2CO_3$ | DMI | 150 | 13 | 67 | 67 |
| 5 | 2 | DBU | DMI | 150 | 10 | 3 | 3 |
| 6 | 2 | $P_1$-tBu-TP | DMI | 150 | 7 | 2 | 1 |
| 7 | 2 | $P_2$-tBu | DMI | 150 | 14 | 61 | 66 |
| 8 | 2 | $P_4$-tBu | DMI | 150 | 16 | 67 | 72 |
| 9 | 2.5 | $P_4$-tBu | DMI | 150 | 7 | 74 | 73 |
| 10 | 2.5 | $P_4$-tBu (5 mol%) | DMI | 150 | 12 | 58 | 58 |
| 11 | 2.5 | $P_4$-tBu | DMI | 120 | 11 | 50 | 45 |
| 12 | 2.5 | $P_4$-tBu | DMI | 100 | 12 | 37 | 31 |
| 13 | 2.5 | $P_4$-tBu | DMAc | 150 | - | 95 | 84 |
| 14 | 2.5 | $P_4$-tBu | DMF | 150 | 3 | 65 | 52 |
| 15 | 2.5 | $P_4$-tBu | PhCN | 150 | 28 | 32 | 12 |
| 16 | 2.5 | $P_4$-tBu | $(EtOCH_2CH_2)_2O$ | 150 | 19 | 10 | 3 |
| 17 | 2.5 | $P_4$-tBu | Xylene | 150 | 19 | 11 | 2 |
| 18 | 2.5 | $P_4$-tBu + $K_3PO_4$ (5 mol%) | DMAc | 150 | - | >99 (85) | >99 (61) |

[a] A mixture of 1 (powder, 0.1 mmol relative to the molecular weight of the monomer), 2a (0.2 mmol for entries 1–8, 0.25 mmol for entries 9–18), catalyst (0.01 mmol), and solvent (0.2 mL) was stirred for 16 h. Yields were determined by [1]H NMR. Numbers in parentheses are isolated yields.

**Fig. 2 | Structure of used organic bases.** These $pK_a$ values in DMSO are shown in parentheses. Bases such as $P_2$-$t$Bu and $P_4$-$t$Bu with high basicity showed catalytic activity.

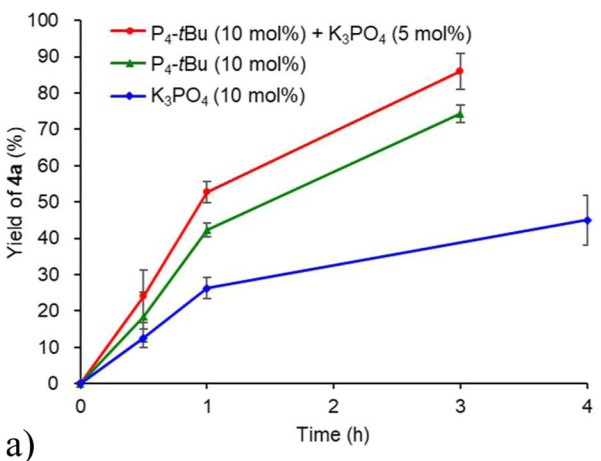

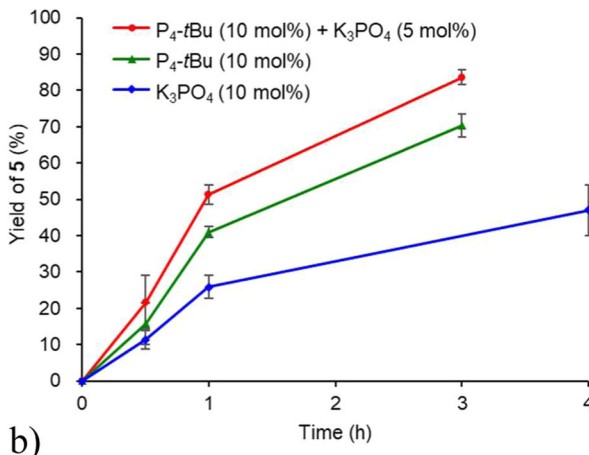

**Fig. 3 | PEEK decomposition time course.** Reaction conditions: PEEK (0.1 mmol relative to the molecular weight of monomer), **2a** (2.5 equiv.) at 150 °C in the presence of $P_4$-$t$Bu (10 mol%) and $K_3PO_4$ (5 mol%) (red line), $P_4$-$t$Bu (10 mol%) (green line), and $K_3PO_4$ (10 mol%) (blue line). **a** Yields of **4a** under various conditions are plotted as the average of the three runs with standard errors. **b** Yields of **5** under various conditions are plotted as the average of the three runs with standard errors.

enhances the nucleophilicity of the counteranions[49–63]. For example, Shigeno, Korenaga, and Kondo recently reported that $P_4$-$t$Bu activates an alkanethiol ($pK_a$ of $n$-BuSH: 17.0 in DMSO)[64]. In this study, highly basic phosphazene bases $P_4$-$t$Bu and $P_2$-$t$Bu ($pK_{BH+}$ of $P_2$-Et: 21.15 in DMSO) exhibited good catalytic activity in comparison with weaker bases such as DBU ($pK_{BH+}$ 13.9 in DMSO) and $P_1$-$t$Bu-TP ($pK_{BH+}$ 17.4 ± 1.2 in DMSO) (Table 1, Entries 5-8). Thus, the basicity and size of the catalysts are important for this reaction to enhance the nucleophilicity of the counter anion. Increasing the amount of **2a** from 2 equiv. to 2.5 equiv. enhanced the yield of **4a** and **5** (Table 1, Entry 9). On the other hand, high loading of $P_4$-$t$Bu (20 mol%) had little effect (see Supplementary Information, Table S1, Entry 11), suggesting that increasing the amount of $P_4$-$t$Bu does not directly lead to an increase in yields of **4a** and **5**. The $P_4$-$t$Bu catalyst loading was successfully reduced to 5 mol%, albeit with slightly decreased yield (Table 1, Entry 10). The reaction at lower temperatures (120 and 100 °C) decreased the yield (Table 1, Entries 11 and 12). As mentioned above, PEEK is insoluble in organic solvents, but previous studies[39,40] showed that solvents affect the reactivity of the decomposition. So, we checked the solvent effects for this decomposition in detail. As a result, $N$,$N$-dimethylacetamide (DMAc) was effective in the conditions whereas other solvent such as $N$,$N$-dimethyl-formamide (DMF), benzonitrile (PhCN), diethylene glycol diethyl ether (($C_2H_5OCH_2CH_2$)$_2$O), and xylene decreased the yield (Table 1, Entries 13-17). $P_4$-$t$Bu dissolves in these solvents so that the decomposition reactivity may be affected by the polarity of the solvents[53]. Finally, we found that the catalyst combination of $P_4$-$t$Bu (10 mol%) and $K_3PO_4$ (5 mol%) enhanced the reactivity of the present decomposition and gave **4a** and **5** in excellent yields in DMAc solvent (Table 1, Entry 18, see Method and section 7-1 in Supplementary Methods).

**Experimental mechanistic studies**

To evaluate the present catalytic decomposition reactivity, we monitored the yields of **4a** and **5** during the reaction of PEEK powder **1** with **2a** catalyzed by

$P_4$-$t$Bu (10 mol%) and $K_3PO_4$ (5 mol%), $P_4$-$t$Bu (10 mol%), and $K_3PO_4$ (10 mol%) (Fig. 3, see section 9-1 and Table S5 in Supplementary Methods). Under the three conditions, **4a** and **5** were formed after 30 minutes. Moreover, high yields of **4a** and **5** were obtained after 3 h under the conditions using $P_4$-$t$Bu and $K_3PO_4$. These observations indicate that the decomposition proceeded rapidly. When $P_4$-$t$Bu catalyst was only used, decomposition proceeded faster than when $K_3PO_4$ catalyst was used. The catalyst combination of $P_4$-$t$Bu and $K_3PO_4$ increased the rate of formation of **4a** and **5** compared to the use of $P_4$-$t$Bu alone. These results indicate that the use of the $P_4$-$t$Bu catalyst allowed for rapid decomposition. The $K_3PO_4$ assisted this catalytic activity of $P_4$-$t$Bu.

To understand the solvent effect for the decomposition of PEEK, we examined the swelling behavior of PEEK resins. PEEK granules or plates were heated in solvents such as DMAc, DMF, PhCN, (EtOCH₂CH₂)₂O, and xylene at 150 °C for 19 h (See section 9-2, Table S6, and Fig. S1 in Supplementary Methods). As a result, these solvents increased the mass of the PEEK granules and plates (105~109 wt%) whereas the resins were apparently unchanged. These observations suggested that the swelling effect of PEEK does not affect the decomposition reactivity. Then we examined the reaction of 4,4'-diphenoxy-benzophenone (**6**) as a PEEK model compound with 2.5 equiv. of **2a** and 10 mol% of $P_4$-$t$Bu at 150 °C for 3 h (Fig. 4a). The reaction using DMAc formed **4a** and phenol in an excellent yield. On the other hand, use of other solvents such as PhCN, ($C_2H_5OCH_2CH_2$)$_2$O, and xylene decreased the yield of **4a**. In these cases, the reactions were not complete even after 22 h. Thus, DMAc as the high polar solvent enhanced the reactivity of the thiolate generated by the combination of the thiol and $P_4$-$t$Bu and probably promoted the cleavage of the carbon–oxygen bonds and the decomposition[45].

Next, we carried out NMR experiments to shed light on the combination of the thiol, $P_4$-$t$Bu, and $K_3PO_4$. The reaction of 4-$tert$-butylphenylthiol (0.02 mmol) and $P_4$-$t$Bu (0.02 mmol) in the presence of $K_3PO_4$ (0.02 mmol) was examined in DMF-$d_7$ (0.5 mL) at 25 °C (see section 9-3 in

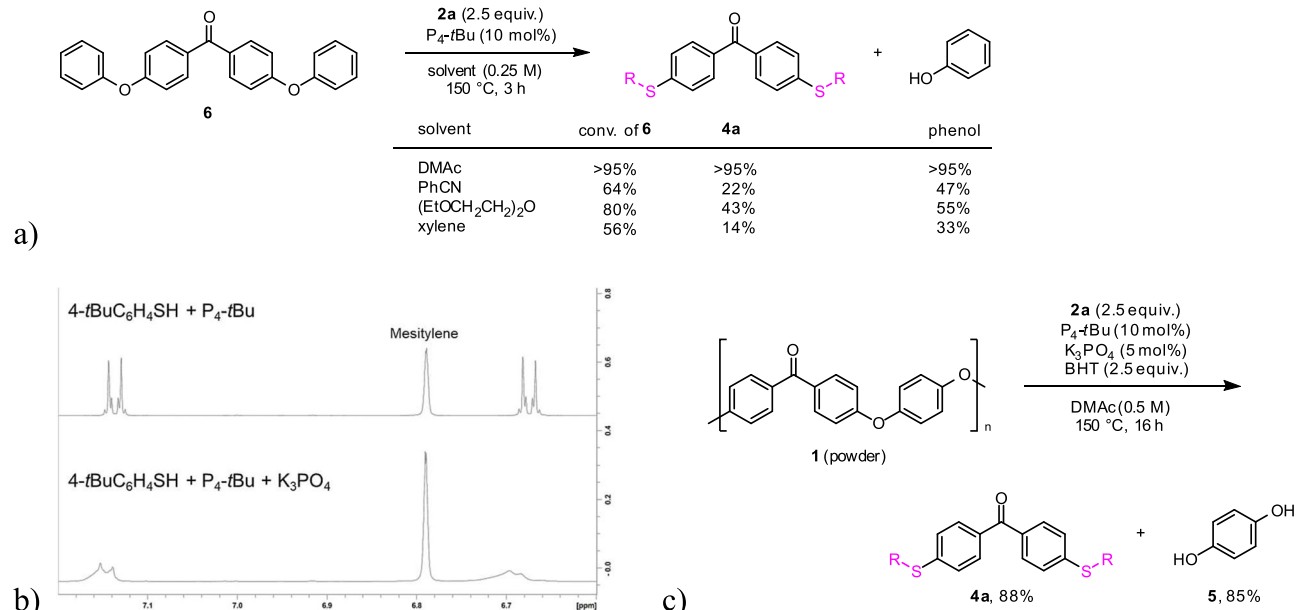

a)

b)

c)

**Fig. 4 | Mechanistic studies. a** Examination of the solvent effect using a model substrate **6** under P$_4$-$t$Bu catalyst. Yields of the products were determined by $^1$H NMR. **b** $^1$H NMR spectra indicating formation of reactive thiolate by reaction of 4-*tert*-butylphenylthiol (0.02 mmol) and P$_4$-$t$Bu (0.02 mmol) in the presence or absence of K$_3$PO$_4$ (0.02 mmol) in DMF-$d_7$. Mesitylene was used as an internal standard. **c** Examination of decomposition of PEEK in the presence of 3,5-di-*tert*-butyl-4-hydroxytoluene (BHT). The radical inhibitor does not affect the decomposition.

were broadened in comparison with the case in the absence of K$_3$PO$_4$ (Fig. 4b). In addition, these signals were different from the combination of the thiol and K$_3$PO$_4$ (see Supplementary Fig. S4). These results indicated that [P$_4$-$t$Bu-H]$^+$·[S(C$_6$H$_4$-$t$Bu)]$^-$ was initially formed and the [S(C$_6$H$_4$-$t$Bu)] anion coordinated to K$_3$PO$_4$ in the equilibrium state. Density functional theory (DFT) calculations suggested that the NBO charge of the phenylthiolate coordinating to K$_3$PO$_4$ is more nucleophilic than the non-coordinating one (see section 9-4 in Supplementary Methods and Supplementary Data 2). We assumed that this catalyst combination activates the thiol for the smooth decomposition of PEEK.

Aromatic nucleophilic substitution with thiolate anions is known to proceed via the S$_N$Ar or S$_{RA}$1 mechanism[65–67]. In the S$_{RA}$1 mechanism, thiyl radicals are thought to be involved. However, this catalytic decomposition of PEEK gives hydroquinone which inhibits the generation of free radicals. We examined the decomposition with **2a** under P$_4$-$t$Bu/K$_3$PO$_4$ catalyst with 3,5-di-*tert*-butyl-4-hydroxytoluene (BHT, 2.5 equiv.), a radical inhibitor, at 150 °C for 16 h and observed the formation of **4a** and **5** in high yields (Fig. 4c). These results ruled out the possibility of a radical pathway for the decomposition. Of note, 2,2,6,6-tetramethylpiperidine 1-oxyl (TEMPO) as a typical radical scavenger was not suitable for this experiment, which converted **2a** into the corresponding disulfide in the absence of PEEK (see section 9-5 and Table S7, S8 in Supplementary Methods)[68].

### Proposed mechanism

A plausible pathway for the chemical decomposition catalyzed by P$_4$-$t$Bu and K$_3$PO$_4$ is shown in Fig. 5. The thiol is initially activated by P$_4$-$t$Bu to form a thiolate that interacts with K$_3$PO$_4$ in the equilibrium state. The sulfur center of the thiolate attacks the *ipso*-carbon bound to oxygen in the benzophenone unit in PEEK to form an anionic intermediate. The aryloxy anion is released to complete carbon–sulfur bond formation. K$_3$PO$_4$ may enhance the reactivity of the thiolate and assist in the release of the aryloxy anion. The generated aryloxy anion activates the thiol to form the organic thiolate and arenols. In fact, the basicity of arenols (p$K_a$ in DMSO of PhOH: 18.0; *p*-MeC$_6$H$_4$OH: 18.9)[69] is higher than that of thiols (p$K_a$ in DMSO of *n*-BuSH: 17.0; PhSH: 10.3)[64]. This series of processes occurs repeatedly to generate the dithiobenzophenone **4** and hydroquinone (**5**).

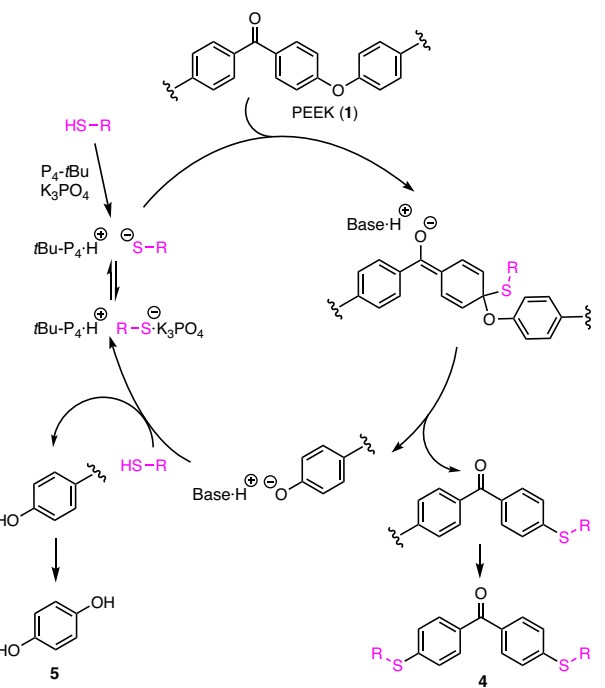

**Fig. 5 | Plausible pathway for chemical decomposition of PEEK.** Cleavage of carbon-oxygen main chains by organo thiolate generated from the reaction of thiol and catalysts.

Supplementary Methods). As a result, a $^{31}$P{$^1$H} NMR spectrum suggested the formation of [P$_4$-$t$Bu-H]$^+$ (Fig. 4b, see Supplementary Fig. S5 compared with Fig. S2) and mass peaks were also observed at *m/z* 634 in ESI-TOF-(+)-MS and *m/z* 165 in ESI-TOF-(-)-MS mass spectra, confirming the generation of [P$_4$-$t$Bu-H]$^+$·[S(C$_6$H$_4$-$t$Bu)]$^-$. The same results were observed in the absence of K$_3$PO$_4$ (see Supplementary Fig. S3). On the other hand, in $^1$H NMR spectrum, the resonances for the aryl doublets (δ 6.69 and δ 7.14)

**Table 2 | Scope of super engineering plastics for the chemical decomposition with 2-ethylhexanethiol in the presence of P$_4$-$t$Bu and K$_3$PO$_4$ in DMAc[a]**

| Entry | Polymer | Products[b] |
|---|---|---|
| 1 | PSU, pellet ($M_w$ 35000, $M_n$ 16000) (**7**) | **8a**, 89%    **9**, 94% |
| 2 | PSU, pellet ($M_w$ 60000) (**7'**) | **8a**, 99%    **9**, >99% |
| 3 | PEES, pellet (-) (**10**) | **8a**, >99%    **5**, 80% |
| 4 | PPSU, powder (-) (**11**) | **8a**, 95%    **12**, >99% |
| 5[b] | PESU, pellet (-) (**13**) | **8a**, 22%    **14**, 21%    **15**, 13% |
| 6 | PEI, pellet (-) (**16**) | **17**, 75%    **9**, >99% |

[a] A mixture of polymer (0.1 mmol relative to the molecular weight of the monomer), **2a** (0.25 mmol), P$_4$-$t$Bu (0.01 mmol), K$_3$PO$_4$ (0.005 mmol) and DMAc (0.2 mL) was stirred for 16 h at 150 °C. Isolated yields are shown. [b] 4-((4-(4-((4-((2-Ethylhexyl)thio)phenyl)sulfonyl)phenoxy)phenyl)sulfonyl)phenol was obtained in 15% yield.

## Substrate scope

With the optimum conditions using both P$_4$-$t$Bu and K$_3$PO$_4$ in hand, we examined the chemical decomposition of other super engineering plastics such as polysulfone (PSU), polyetherethersulfone (PEES), polyphenylsulfone (PPSU), polyethersulfone (PESU), and polyetherimide (PEI) which were analyzed by high-temperature GPC analysis prior to use (see section 9-6 and Table S9 in Supplementary Methods). These resins have cleavable aryl-oxygen bonds affected by electron-withdrawing groups in a manner similar to PEEK. PSU is composed of diphenylsulfone and bisphenol A. Since thiolate anions can cleave aryl-SO$_2$ bonds[70–74], we were concerned that the present catalytic method may cleave the aryl-SO$_2$ bond in the diphenylsulfone unit as well as the target C–O main chain. However, we found that polysulfone (PSU) pellets **7** (purchased from Sigma-Aldrich) and **7'** (purchased from Acros Organics) with different $M_w$ ($M_w$ 35000 and $M_w$ 60000) in each of the catalog specifications underwent the decomposition with **2a** via selective C–O bond cleavage[38] to furnish the corresponding 4,4'-dialkylthiobenzosulfone (**8a**) and bisphenol A (**9**) in high yields (Table 2, Entries 1 and 2). In addition, there was no clear difference in the reaction rate between **7** and **7'** (see Supplementary Table S2). In the same way, PEES pellets (**10**) or PPSU powder (**11**) could be converted into **8a** and hydroquinone (**5**) or 4,4'-dihydroxybiphenyl (**12**) in high yields (Table 2, Entries 3 and 4). In the case of PESU (**13**) consisting of repeating oxy-diphenylsulfone units, three products **8a**, 4-alkylthio-4'-hydroxy-diphenylsulfone **14**, and bisphenol S (**15**) were obtained (Table 2, Entry 5). PEI is composed of repeating structures of phenylene-1,3-bisphthalimide and bisphenol A. In this case, imide bonds in the phthalimide units may be cleaved by sulfur nucleophiles[75]. Nevertheless, the C–O main chains were successfully cleaved selectively in the decomposition of PEI pellets **16** with **2a**, giving dithiofunctionalized phenylene-1,3-bisphthalimide **17** and **9** in good yields (Table 2, Entry 6).

We then explored the scope of thiols under the catalytic decomposition of PSU pellets **7** (Fig. 6). 2-Phenylethanethiol or 2-mercaptoethanol underwent decomposition at 100 °C to form **8b** (see section 7-2 in Supplementary Methods) or **8c** and bisphenol A (**9**) in good yields. Triethoxysilyl-substituted propanethiol and cyclopentanethiol were used in the decomposition and the corresponding decomposition products **8d** and **8e** were obtained. Trimethylsilylmethylthiol gave 4,4'-dimethylthiodiphenylsulfone **8f** and **9** in high yields via desilylation. Not only alkanethiols but also 4-$tert$-butylbenzenethiol could be utilized for decomposition with only NaO$t$Bu catalyst (20 mol%) to form the corresponding monomer **8g** in 98% yield together with **9** quantitatively (see Supplementary Table S3 and section 7-3 in Supplementary Methods)). Instead of PSU, we attempted the decomposition of PEEK powder with 4-$tert$-butylbenzenethiol under the P$_4$-$t$Bu/K$_3$PO$_4$ or NaO$t$Bu catalyst in DMAc but the yield of the product, 4,4'-di(arylthio)benzophenone **4b**, was low (see Supplementary Table S4 and section 7-4 in Supplementary Methods). At that time, a suspension containing precipitated **4b** and its intermediates were obtained. Considering that the poor solubility of the products may have decreased the reactivity, we modified the conditions using a P$_4$-$t$Bu/Cs$_2$CO$_3$ catalytic combination in DMI to enhance the solubility. As a result, **4b** was obtained in high yield, albeit with a long reaction time.

## Utility of the decomposition method

To demonstrate the scalability of the decomposition method, a gram-scale reaction of PSU pellets (**7**) with cyclopentanethiol catalyzed by 5 mol% of P$_4$-$t$Bu and K$_3$PO$_4$ was carried out. The desired products **8e** and **9** were isolated in 78% and 75% yields, respectively (Fig. 7a, see section 7-5 in Supplementary Methods). It is worth noting that this catalytic method was applicable to composite materials. Shaved powder of 30 wt% carbon-fiber reinforced PEEK (**1'**) underwent the decomposition with **2a** to form **4a** and

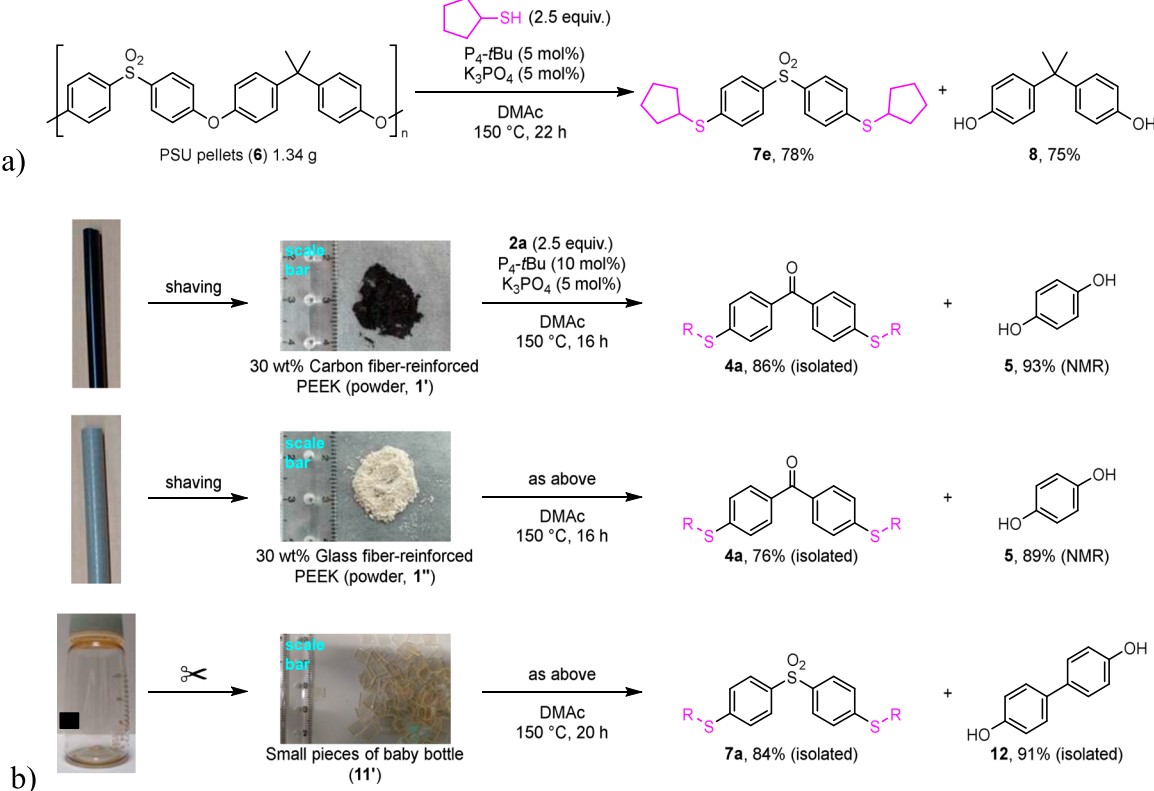

**Fig. 6 | Scope of thiols for the chemical decomposition of PSU.** Reaction conditions: a mixture of polymer (0.1 mmol relative to the molecular weight of monomer), thiol (0.25 mmol), P$_4$-$t$Bu (0.01 mmol), K$_3$PO$_4$ (0.005 mmol), and DMAc (0.2 mL) was stirred for the time shown at 150 °C. Isolated yields are reported. [a] NaO$t$Bu (0.02 mmol) only was used as the catalyst. [b] Cs$_2$CO$_3$ (0.01 mmol) and DMI (0.2 mL) were used instead of K$_3$PO$_4$ and DMAc.

**Fig. 7 | Utility of this decomposition method. a** Gram-scale decomposition of PSU pellets (1.34 g) with cyclopentane thiol under P$_4$-$t$Bu (5 mol%) and K$_3$PO$_4$ (5 mol%) catalysts in DMAc at 150 ºC. **b** Decomposition of 30% carbon fiber- or 30% glass fiber-reinforced PEEK or a baby bottle made up of PPSU with 2-ethylhexanethiol under the catalytic conditions using P$_4$-$t$Bu (10 mol%) and K$_3$PO$_4$ (5 mol%) in DMAc at 150 ºC.

**5** in good yields comparable to those obtained from neat PEEK powder (Fig. 7b, see section 7-6 in Supplementary Methods). 30 wt% Glass-fiber reinforced PEEK (**1'''**) was converted into **4a** and **5** in the same way. In addition, small pieces of a baby bottle made up of PPSU (**11'**) as a representative consumer resin were transformed into products, **8a** and **12**, in high yields (see section 7-7 in Supplementary Methods).

## Utility of products

Sulfur functional groups in the products can be utilized in various transformations to yield functional molecules. For example, **8e** was applicable to the double cross-coupling with 4-decylaniline under palladium-catalyzed conditions[76] to give the corresponding double amination product **18** (Fig. 8a, see section 7-8 in Supplementary Methods). Double phenylation of **4b** using diphenyliodonium trifluoromethanesulfonate and copper acetate catalyst in 1,2-dichloroethane at 100 °C, based on a reported method[77], gave benzophenone 4,4′-bis (diarylsulfonium) salt **19** in excellent yield (Fig. 8b, see section 7-9 in Supplementary Methods). Such sulfonium groups are more reactive leaving groups than their parent sulfur functional groups. Thus, the sulfonium groups in **19** could be converted into fluorine by potassium

**Fig. 8 | Functionalization of products. a** Cross-coupling of **8e** with 4-decylaniline to give 4,4'-sulfonylbis(*N*-(4-decylphenyl)aniline) (**18**). **b** Conversion of **4b** into benzophenone-based disulfonium salt **19** followed by fluorination to form 4,4'-difluorobenzophenone (**20**). **c** Conversion of **8g** into diphenylsulfone-based disulfonium salt **21** followed by fluorination to form di(4-fluorophenyl)sulfone (**22**) or etherification to form 4,4'-bis(aryloxy)diphenylsulfone **23**. SingaCycle-A1: Chloro[[1,3-bis(2,6-diisopropylphenyl)imidazol-2-ylidene](*N*,*N*-dimethylbenzylamine)palladium(II)]. DCE: 1,2-dichloroethane. Kryptofix® 222: 4,7,13,16,21,24-hexaoxa-1,10-diazabicyclo[8.8.8]hexacosane. DMF: *N*,*N*-dimethylformamide. *p*-Anisyl: 4-methoxyphenyl.

fluoride and Kryptofix® 222 (4,7,13,16,21,24-hexaoxa-1,10-diazabicyclo[8.8.8]hexacosane) in *N*,*N*-dimethylformamide at 60 °C (see section 7-10 in Supplementary Methods)[78]. Of note, the product, 4,4'-difluorobenzophenone (**20**), is used as a monomer for PEEK[79,80]. PSU-depolymerized product **8g** was also applicable to this transformation sequence. Double phenylation of **8g** afforded diphenylsulfone 4,4'-bis(diarylsulfonium) salt **21** (Fig. 8c, see section 7-11 in Supplementary Methods). Subsequent fluorination of **21** gave bis(4-fluorophenyl)sulfone (**22**) in 87% yield (see section 7-12 in Supplementary Methods), which is a monomer of diphenylsulfone-based polymers such as PSU[81,82], PPSU[83–90], PESU[91,92], and PEES[93]. In addition, **21** reacted with *p*-methoxyphenol in the presence of Cs$_2$CO$_3$ to give 4,4'-bis(*p*-anisyloxy)diphenylsulfone (**23**) in 79% yield (see section 7-13 in Supplementary Methods).

## Conclusion

In this study, we demonstrated that the depolymerization-like chemical decomposition of robust super engineering plastics such as PEEK, PSU, PEES, PPSU, PESU, and PEI occurred smoothly with thiols at moderate temperature under the catalytic combination of bulky organic super bases, P$_4$-*t*Bu, and inorganic bases, K$_3$PO$_4$ or Cs$_2$CO$_3$. DMAc solvent also promoted the carbon-oxygen bond cleavages in a low-weight molecule and insoluble PEEK due to its polarity under the conditions. Various thiols were applied to this decomposition to afford monomer-like thiofunctionalized arenes and bisphenols in high yields. In addition, carbon fiber- or glass fiber-reinforced resins and a baby bottle made of PPSU as a representative consumer resin material were utilized in this catalytic decomposition. From a synthetic perspective, thiofunctionalities in the arene products act as leaving groups and can be transformed into various substituents such as amino groups and fluorine. Notably, fluorinated arenes are parent monomers for synthesizing super engineering plastics. This shows that the present catalytic decomposition method can be utilized not only for chemical recycling but also for upcycling. This development will expand the decomposition of other robust polymer materials with various reagents under this effective catalytic system.

## Methods

### General procedure for catalytic chemical decomposition of PEEK

To a mixture of PEEK powder (28.8 mg, 0.100 mmol relative to the molecular weight of the monomer), and potassium phosphate tribasic (1.1 mg, 0.0050 mmol) was added *N*,*N*-dimethylacetamide (0.20 mL), P$_4$-*t*Bu phosphazene base in hexane solution (1-*tert*-butyl-4,4,4-tris(dimethylamino)-2,2-bis[tris(dimethylamino)-phosphoranylidenamino]-2λ5,4λ5-catenadi(phosphazene), 0.8 M, 0.0125 mL, 0.010 mmol), and 2-ethylhexanethiol (36.6 mg, 0.250 mmol) in a 3 mL vial under argon atmosphere. The resultant mixture was stirred at 150 °C for 16 h. The reaction mixture was cooled to room temperature. The mixture was analyzed by [1]H NMR in acetone-*d$_6$* to determine the yields of the products, **4a** and hydroquinone (**5**), using 1,4-dioxane as an internal standard. The reaction mixture was concentrated in vacuo. The crude product was purified by column chromatography on silica gel (hexane/ethyl acetate 96:4 to 7:3) to give bis(4-(2-ethylhexylthio)phenyl)methanone (85%, 39.9 mg) and 1,4-hydroquinone (61%, 6.7 mg).

**General information**. See Supplementary Methods, general information (page S3).

**Chemicals**. See Supplementary Methods, chemicals (page S3).

**NMR charts**. See Supplementary Data 1, NMR spectra of obtained chemicals.

## Data availability

The data obtained in this study are available within this article and its supplementary information and are also from the corresponding authors upon reasonable request. Original [1]H and [13]C spectra of the compounds

obtained in this manuscript are available in Supplementary Data 1. The computed energy values and coordinates are available in Supplementary Data. 2.

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

## Acknowledgements

This work was supported financially by PRESTO (JPMJPR21N9 to Y.M.) from the JST, Iketani Science and Technology Foundation, Tobe Maki Scholarship Foundation, Grants-in-Aid for Scientific Research (C) (19K05481 to Y.M.) from the JSPS, and Department of Materials and Chemistry, AIST. Y.M., N.M., and Y.N. also acknowledge the DIC Corporation. Y.M. would like to thank Dr. Masanori Shigeno for discussions about the catalytic activity. Y.M would like to thank JST, ERATO (JPMJER2103), and Prof. Kyoko Nozaki and her lab members, Prof. Kohei Takahashi, Prof. Takanori Iwasaki, Prof. Shuhei Kusumoto, and Prof. Xiongjie Jin, for their discussions on this project. We would like to thank Ms. Risa Kawato for her assistance in the high-temperature GPC analysis.

## Author contributions

Y.M. conceived the idea and designed the whole experiment with S.I. and N.M. Y.M., S.I., and N.M. performed the experiments. Y.M., S.I., and N.M. contributed to writing the manuscript and participated in data analyses and discussions. S.I. and N.M. contributed equally to this paper. Y.M. performed the DFT calculations. Y.N. and M.Y. supported this project. Y.M. revised the paper.

## Competing interests

The authors declare no competing interests.
