## [Peer Review File · Communications Chemistry]

REVIEWERS' COMMENTS:

Reviewer #1 (Remarks to the Author):

The manuscript describes degradation of various oxyphenylene type engineering plastics using thiol as a reagent with catalyst such as P4-tBu K3PO4. The method is original of the authors and the results are very worth considering chemical recycles of engineering plastics, which is one of the most difficult issues. I recommend that this paper should be published in Communications Chemistry. However, there are some points that need to be clarified, as follows.

Terminology

The authors use the word of “depolymerization” in this study. However, 2.5 equivalent (more than catalytic amount) of thiol were used to decompose 1 and other polymers, and the thiol unit was incorporated in the product. Is it reasonable to categorize this reaction in “depolymerization”? Please make a clear statement in the manuscript.

Figure 3

Yields of 4a and 5 in the presence of “P4-tBu (10 mol%) + K3PO4 (5 mol%)” are smaller than the sum of those in the presence of “P4-tBu (10 mol%)” and “K3PO4 (5 mol%)”. The referee thinks the increase in the yields of 4a and 5 in the presence of “P4-tBu (10 mol%) + K3PO4 (5 mol%)” can be simply explained by the increase in the total amount of the catalyst from 10 mol% to 15 mol%. The authors should mention about this point in the manuscript.

Minor comments

Page 4, Line 72 Typo, “Mm ~ 10300”.

Reviewer #2 (Remarks to the Author):

In this work, it reports the catalytic depolymerization of oxyphenylene-based super engineering plastics such as polyetheretherketone, polysulfone, and polyetherimide using thiols via selective carbon–oxygen main chain cleavage to form monomer-type molecules, electron-deficient arenes with sulfur functional groups and bisphenols. However, this work is similar to the previous work both in terms of catalysts and reactions (Commun. Chem. 6, 14 (2023)). It is less innovative and more of an extension of previous work. Therefore, I do not think this article is suitable for publication in this journal.

In the meantime, I expect the following questions to be resolved before submitting to other journals.

1. In the second paragraph of the introduction, I do not understand the logical relationship between persistent resins and super engineering plastics. So I think the introduction should be carefully revised.
2. The paper lacks a story and is structured more like a report. I propose to adjust the structure of the paper to make it more logical.
3. Table 1 shows whether other catalysts show good activity in DMAC. I guess the different solubility of the catalyst in different solvents will affect the reaction activity. This needs further confirmation to make the study more rigorous.

Point-by-Point response to referees

All the responses made are summarized in followings.

Comments from Reviewer 1:

1-1) Terminology. The authors use the word of “depolymerization” in this study. However, 2.5 equivalent (more than catalytic amount) of thiol were used to decompose **1** and other polymers, and the thiol unit was incorporated in the product. Is it reasonable to categorize this reaction in “depolymerization” ? Please make a clear statement in the manuscript.

Response to 1-1)

We appreciate the Reviewer 1 for the valuable comment. As mentioned by Reviewer 1, we used 2.5 equiv. of thiols. However, the thiol was used as the reagent for the depolymerization of PEEK and other super engineering plastics to form two products: dithiofunctionalized monomer type products and bisphenols as the monomers. In this respect, we thought of this transformation as the depolymerization. When stable thiols such as 2-ethylhexylthiol and cyclopentylthiol are used, these are basically considered possible to recover them after the reaction.

1-2) About Figure 3. Yields of **4a** and **5** in the presence of “P4-*t*Bu (10 mol%) + K3PO4 (5 mol%)” are smaller than the sum of those in the presence of “P4-*t*Bu (10 mol%)” and “K3PO4 (5 mol%)”. The referee thinks the increase in the yields of **4a** and **5** in the presence of “P4-*t*Bu (10 mol%) + K3PO4 (5 mol%)” can be simply explained by the increase in the total amount of the catalyst from 10 mol% to 15 mol%. The authors should mention about this point in the manuscript.

Response to 1-2)

We appreciate the Reviewer 1 for the valuable comment. Although 20 mol% of P4-*t*Bu was used in the depolymerization with 2 equiv. of 2-ethylhexylthiol in DMI, the yields of **4a** and **5** were hardly improved. This result was newly added to Table S1, Entry 11 in the supplementary information. This result suggested that simply increasing the amount of P4-*t*Bu catalyst does not directly lead to an increase in yields of **4a** and **5**. Thus, we revised the corresponding sentence in the manuscript, page 4, line 15-17.

1-3) Minor comments. Page 4, Line 72 Typo, “Mm ~ 10300”

Response to 1-3)

We appreciate the Reviewer 1 for the valuable comment. We corrected this word to “ M_n ~10300”.

Comments from Reviewer 2:

2-1) In the second paragraph of the introduction, I do not understand the logical relationship between persistent resins and super engineering plastics. So I think the introduction should be carefully revised.

Response to 2-1)

We appreciate the Reviewer 2 for the valuable comment. As mentioned by the Reviewer 2, we agreed that the second paragraph in page 2 in the manuscript, which you specifically mentioned in the introduction, needed to be revised. Based on the comments from Reviewer 2, we revised the introduction, second paragraph to show the relationship.

2-2) The paper lacks a story and is structured more like a report. I propose to adjust the structure of the paper to make it more logical.

Response to 2-2)

We appreciate the Reviewer 2 for the valuable comment. We think that the key points of this depolymerization are the depolymerization reactivity, the catalyst and solvent effects, as well as the selectivity toward the functional groups such as sulfone group in PSU, PEES, PPSU, and PESU. According to the reviewer 2 comment 2-2, we revised the above corresponding points to show the contents of the depolymerization in more detail. First, we replaced the word “sulfur nucleophile” to “highly nucleophilic thiolate reagents” in page 2, line 31 to explain the effectiveness of the sulfur reagents. In addition, we newly added the sentence “Under the three conditions, **4a** and **5** were formed after 30 minutes. Moreover, high yields of **4a** and **5** were obtained after 3 h under the conditions using P₄-*t*Bu and K₃PO₄. These observations indicate that the depolymerization proceeded rapidly.” in page 5, line 4-6 to mention the good reactivity. Second, we revised the explanation about the amount of the catalyst (the above responses to 1-2). Third, we performed additional experiments using other solvents such as NMP, DMF, and diethylene glycol diethyl ether under various conditions. These results were newly added to Table 1, Entries 14 and 16 in the manuscript, and Table S1, Entries 19, 20, 22, 27, and 28. Based on previous listed results and the new results using various solvents, we revised the sentence explaining the effect of solvent in the manuscript, page 4, line 19-25. In this relation, we newly added the reference 51. In addition, to address the selectivity of C-O main chain cleavage in preference to C-SO₂ bond in PSU, we added the reference number 38 in the sentence, page 7, line 12. We think that these revisions made the text more logical than the previous version.

2-3) Table 1 shows whether other catalysts show good activity in DMAC. I guess the different solubility of the catalyst in different solvents will affect the reaction activity. This needs further confirmation to make the study more rigorous.

Response to 2-3)

We appreciate the Reviewer 2 for the valuable comment. According to the reviewer 2 comment 2-3, we performed the additional experiments using other solvent to check the reactivity differences (the above responses to 2-2). As a result, DMAC solvent gave the best results as expected. Previous reports showed that P₄-*t*Bu and the combination between P₄-*t*Bu and thiols are soluble in various organic solvents. In addition, PEEK is basically insoluble in organic solvents. These points suggest that the solvents affect reactivity toward PEEK by their polarity, maybe not by their solubility in the catalyst. To address this point, we revised the sentence in the manuscript, page 4, line 21-25, and newly added the reference 51.

REVIEWERS' COMMENTS:

Reviewer #2 (Remarks to the Author):

The authors have made efforts to revise the manuscript and related issues have also been corrected, so I recommend this paper for publication after a minor revision. The detailed comments are as follows:

1. Although the author has revised the introduction, I think that the logical relationship between persistent resins and super engineering plastics should be adjusted to the beginning of the introduction.
2. Recent plastics catalytic conversion progress is missing.

ACS Catalysis, 2023, 13, 3575-3590

Molecular Catalysis, 2023, 542, 113128

3. The word of “depolymerization” is not clearly explained. Please make a clear statement in the manuscript.

Point-by-Point response to referees

All the responses made are summarized in followings.

Comments from Reviewer #2:

1-1) Although the author has revised the introduction, I think that the logical relationship between persistent resins and super engineering plastics should be adjusted to the beginning of the introduction.

Response to 1-1)

We appreciate Reviewer #2 for the valuable comment. As mentioned by Reviewer #2, we revised the abstract and the introduction. Many super engineering plastics are composed of robust aromatic ethers, which are difficult to cleave. This point and many other chemical stability properties such as heat resistance, chemical resistance, and low solubility are considered to distinguish them from other stable resins. From the points of view, we revised the abstract and the 1st and 2nd paragraph in the introduction section. For example, the sentences from the middle to the last part of the 1st paragraph in the abstract were revised. The following sentence was added to the last part in this paragraph “In addition, many super engineering plastics are composed of robust aromatic ethers, which are difficult to cleave.”.

1-2) Recent plastics catalytic conversion progress is missing.

ACS Catalysis, 2023, 13, 3575-3590.

Molecular Catalysis, 2023, 542, 113128.

Response to 1-2)

We appreciate the Reviewer #2 for the valuable comment. These references were newly added in the references 14 and 20.

1-3) The word of “depolymerization” is not clearly explained. Please make a clear statement in the manuscript.

Response to 1-3)

We appreciate the Reviewer #2 for the valuable comment. The editors also mentioned this point. According to the advice from the editors, the word “depolymerization” in the manuscript and the supplementary information was fully replaced to “chemical decomposition” or “depolymerization-like chemical decomposition”.